# Meningeal Sporotrichosis Due to *Sporothrix brasiliensis*: A 21-Year Cohort Study from a Brazilian Reference Center

**DOI:** 10.3390/jof9010017

**Published:** 2022-12-22

**Authors:** Marco A. Lima, Dayvison F. S. Freitas, Raquel V. C. Oliveira, Vivian Fichman, Andréa G. Varon, Andréa D. Freitas, Cristiane C. Lamas, Hugo B. Andrade, Valdilea G. Veloso, Rodrigo Almeida-Paes, Fernando Almeida-Silva, Rosely Maria Zancopé-Oliveira, Priscila M. de Macedo, Antonio C. F. Valle, Marcus T. T. Silva, Abelardo Q. C. Araújo, Maria C. Gutierrez-Galhardo

**Affiliations:** 1Evandro Chagas National Institute of Infectious Diseases (INI), FIOCRUZ, Brazilian Ministry of Health, Rio de Janeiro 21040-900, Brazil; 2Neurology Section, Hospital Universitário Clementino Fraga Filho, The Federal University of Rio de Janeiro (UFRJ), Rio de Janeiro 21941-617, Brazil; 3Deolindo Couto Institute of Neurology (INDC), The Federal University of Rio de Janeiro (UFRJ), Rio de Janeiro 22290-140, Brazil

**Keywords:** *Sporothrix brasiliensis*, sporotrichosis, meningitis, HIV, immune reconstitution inflammatory syndrome

## Abstract

Meningeal sporotrichosis is rare and occurs predominantly in immunosuppressed individuals. This retrospective study explored clinical and laboratory characteristics, treatment, and prognosis of patients with disseminated sporotrichosis who underwent lumbar puncture (LP) at a Brazilian reference center from 1999 to 2020. Kaplan–Meier and Cox regression models were used to estimate overall survival and hazard ratios. Among 57 enrolled patients, 17 had meningitis. Fifteen (88.2%) had HIV infection, and in 6 of them, neurological manifestations occurred because of the immune reconstitution inflammatory syndrome (IRIS). The most frequent symptom was headache (88.2%). Meningeal symptoms at first LP were absent in 7/17 (41.2%) patients. *Sporothrix* was diagnosed in cerebrospinal fluid either by culture or by polymerase chain reaction in seven and four patients, respectively. All but one patient received prolonged courses of amphotericin B formulations, and seven received posaconazole, but relapses were frequent. Lethality among patients with meningitis was 64.7%, with a higher chance of death compared to those without meningitis (HR = 3.87; IC95% = 1.23;12.17). Meningeal sporotrichosis occurs mostly in people with HIV and can be associated with IRIS. Screening LP is indicated in patients with disseminated disease despite the absence of neurological complaints. Meningitis is associated with poor prognosis, and better treatment strategies are needed.

## 1. Introduction

Sporotrichosis is a mycosis caused by dimorphic fungi of the *Sporothrix* genus, especially by the clinically relevant species *Sporothrix schenckii, Sporothrix brasiliensis,* and *Sporothrix globosa* [1]. *S. brasiliensis*, the most virulent among them, is associated with transmission from cats to humans by inoculation (scratches or bites) and is endemic in South America [2]. Clinical manifestations are determined by host immune status and the inoculum load and depth [3]. While most patients develop a localized cutaneous disease, disseminated cutaneous or systemic disease affecting bones and joints, lung, and the central nervous system (CNS) may occur in individuals with immunosuppression (e.g., HIV infection, transplantation, neoplasms, alcoholism, diabetes, and chronic steroid use) [4]. 

The first report of CNS sporotrichosis was in 1957 [5] and, since, data related to clinical evolution and response to treatment are based on case reports and a small series of cases [6,7,8,9,10]. The most frequent neurological syndrome associated with sporotrichosis is chronic meningitis. Freitas et al. described fungal isolation from the cerebrospinal fluid (CSF) of 14.3% of 28 people living with HIV (PLHIV) with disseminated sporotrichosis due to *S. brasiliensis* [7]. Diagnosis is challenging because *Sporothrix* spp. isolation in the CSF is difficult. Complications such as hydrocephalus and vascular lesions are frequent, and meningitis due to *Sporothrix* sp. is associated with high mortality [8]. Important questions related to the treatment and follow-up of these patients remain unanswered. 

Herein, we describe the clinical and demographical aspects of 57 patients with disseminated sporotrichosis who underwent lumbar puncture to investigate meningeal dissemination and the neurological manifestations, radiological and CSF characteristics, treatment, and prognosis of these patients.

## 2. Materials and Methods

This is a retrospective cohort study enrolling patients with disseminated cutaneous sporotrichosis associated with immunosuppression and disseminated forms of this mycosis who underwent lumbar puncture at the Evandro Chagas National Institute of Infectious Diseases (INI)/Oswaldo Cruz Foundation (FIOCRUZ), Rio de Janeiro, Brazil, from March 1999 to March 2020. This study was approved by the Institutional Research Ethical Committee (#88551018.9.0000.5262). All patients received confirmation of sporotrichosis by the culture of clinical specimens from skin lesions or other sites. 

Clinical specimens were cultured on BBL Sabouraud Dextrose Agar 2% (Becton Dickinson Co., Sparks, MA, USA) and Mycosel Agar (Becton Dickinson Co., Sparks, MA, USA) incubated at 25 °C for up to four weeks. Dimorphism of suspected *Sporothrix* spp. colonies was assessed on Brain Heart Infusion Agar (Becton Dickinson Co., Sparks, MA, USA) at 35 °C for seven days. Molecular identification of pure and viable *Sporothrix* spp. strains isolated from clinical samples was performed using a species-specific polymerase chain reaction (PCR) protocol [11]. 

Meningitis was defined as *Sporothrix* spp. isolation from the CSF or as the detection of biochemical or cytological alterations in the CSF in patients with cutaneous sporotrichosis. Patients who underwent lumbar puncture due to other causes or who had other confirmed CNS diagnoses with no detection of *Sporothrix* sp. in CSF were excluded. The institutional routine included a search for VDRL test, staining and cultures for mycobacteria, bacteria, and other fungi.

Data collected from medical charts included demographic and epidemiological data. Clinical information was the type of neurological complaints (headache, vomiting, motor deficits, ataxia, altered level of consciousness, nuchal rigidity, cranial nerve palsy, seizure, and cognitive dysfunction), presence of immunosuppression, and for PLHIV, time from HIV infection diagnosis, duration of treatment, and development of immune reconstitution inflammatory syndrome (IRIS). The criteria for IRIS in the CNS was the appearance and/or worsening of meningeal sporotrichosis after immunological improvement (increase in CD4+ T lymphocyte count or and more than 1 log decrease in HIV viral load) due to the initiation of combination antiretroviral therapy (ART)). Laboratory data included CSF parameters (cells, proteins, and glucose levels), detection of *Sporothrix* spp. In culture of the CSF (five patients also had CSF tested in a study involving PCR for *Sporothrix* spp., as previously reported) [12], imaging characteristics (presence of meningeal contrast enhancement, hydrocephalus, and ischemic or hemorrhagic cerebral lesions) and in PLHIV, CD4+ T lymphocyte count, nadir CD4+ T lymphocyte count, and HIV viral load. Treatment and outcome data were the treatment regimen, number of amphotericin B (AMB) days for treatment of meningitis, use of posaconazole and steroids (prednisone ≥ 20 mg/day or equivalent dosage of other steroid for more than 7 days), admission to intensive care unit, number of admissions for treatment of meningitis relapses, and death.

Patients with disseminated sporotrichosis who developed neurological symptoms and signs of meningitis during follow-up were defined as Group 1. Patients with disseminated sporotrichosis, no neurological symptoms, and CSF cell count ≤ 5 cells/mm³ were defined as Group 2. 

Summary measures (mean, minimum, and maximum) were used for quantitative variables and frequencies (counts and percentages) were used for qualitative variables. Survival analysis was used to investigate the outcome time to death (overall survival). The survival time was calculated by the difference between the date of diagnosis and the outcome. Censoring was defined by the date of the hospital discharge or last visit. The Kaplan–Meier method was employed to analyze the overall survival and we show the median of survival time. Semi-parametric Cox regression models were used to estimate the effects on the overall survival, using time as a counting process. First, single covariate Cox models were used to estimate the Crude Hazard Ratios (Crude HR); then, a multiple covariate Cox model was used to adjust the effect of meningitis for confounders (sex, skin color/ethnicity, age, onset time, and HIV status), named Adjusted HR. Schoenfeld residuals did not reject the proportionality assumption of the final multiple covariate model.

A subgroup analysis was performed in the patients with meningitis (Group 1) to investigate the factors associated with time to develop meningitis. However, due to the low number of cases in Group 1, we describe the individual data of them, avoiding statistical testing. 

Statistical analysis was performed using the software R (version 4.0). The significance level was 0.05.

## 3. Results

### 3.1. Patients with Disseminated Sporotrichosis (Groups 1 and 2)

Fifty-seven patients underwent lumbar puncture during the study period. Four patients who had CSF abnormalities were excluded from the analysis. Two patients had 7 and 8 lymphocytes/mm³ and did not develop clinical manifestations of meningitis during the course of sporotrichosis, and two patients were lost to follow-up.

Among the remaining 53 patients, there were 44 (83%) men, and the mean age was 40.5 years (range 16–72 years). Most patients were non-white (40; 75.5%), and 40 (75.5%) reported close contact with cats. All patients presented sporotrichosis cutaneous lesions (Figure 1). *Sporothrix brasiliensis* was the sole species identified in all the available isolates from any clinical specimen in 35 (66%). 

HIV infection was the most prevalent cause of immunosuppression, being present in 42 (79.2%) participants. Among the 53 patients, 17 showed neurological symptoms or signs of meningitis (Group 1) while 36 did not (Group 2). Epidemiological, clinical, and laboratory findings in the two groups are described in Table 1.

From diagnosis until death (overall survival), the median survival time was 1383 days (Figure 2). Table 2 shows the results of the Crude and Adjusted HR for overall survival. Patients who developed meningitis had a significantly higher chance of death (Adjusted HR = 3.87, IC95% = 1.23;12.17) than patients with disseminated sporotrichosis and no neurological complaints, controlled by sex, skin color, onset time, and HIV status. 

### 3.2. Patients with CNS Sporotrichosis (Group 1)

In the 17 patients with disseminated sporotrichosis and neurological manifestations (Table 3), the mean age at onset was 35.3 years (range 20–57 years), and 15 (88.2%) were men. All but one had non-white skin color. The duration of systemic manifestations of sporotrichosis until lumbar puncture was 6.9 months (range 1–20 months), while the mean duration of neurological symptoms until diagnosis was 2.7 months (range 15 days–15 months) in 10/17 patients. Seven (41.2%) patients did not have meningeal symptoms at the time of first lumbar puncture and developed neurological symptoms after 15 to 850 days (mean 220 days, SD ± 295.7).

Fifteen (88.2%) patients had HIV infection as the predisposing condition. The median CD4+ T lymphocyte count was 110 cells/mm^3^ (range 8–704 cells/mm^3^), and the median duration of HIV diagnosis was 6 months (range 0–17 years). In 9 out of 15 patients, HIV infection was diagnosed as concomitant to disseminated sporotrichosis, and in seven, neurological symptoms developed after the institution of ART, suggesting IRIS. The remaining two patients were not PLHIV; one had leprosy reaction with chronic steroid use, and the other was an alcoholic patient.

The most frequent neurological manifestation at presentation was headache (88.2%), followed by vomiting (64.7%), drowsiness (58.8%), nuchal rigidity (41.2%), cognitive changes (41.2%), seizures (29.4%), cranial nerve palsies (17.6%), and motor deficits (11.8%).

Mean CSF cell count at first lumbar puncture was 71.5/mm^3^ (range 3–363/mm^3^) with lymphocytic predominance in all but one patient, protein level of 117.9 mg/dL (range 33–307 mg/dL), and glucose level of 39.6 mg/dL (range 16–65 mg/dL). In 11/17 (64.7%) patients, *Sporothrix* spp. was identified in the CSF: isolated from CSF culture in seven (41.2%) patients, and a positive PCR for the fungus was present in other four (23.5%) patients, as previously described [12]. *Sporothrix brasiliensis* was the identified species in all the available isolates from any clinical specimen of 11 (64.7%) patients, including five CSF isolates. All patients underwent brain CT scans or MRI. Meningeal contrast enhancement or hydrocephalus were each present in 9 (52.9%) patients (Figure 3A). One patient had a hemorrhagic stroke (Figure 3B), and ischemic lesions were present in two (Figure 3C). In 5 (29.4%) patients, imaging studies disclosed no abnormalities.

On average, all patients but one were treated with different intravenous amphotericin B AMB formulations for 259 days (range 34–645 days). Six (35.2%) patients received posaconazole associated with AMB (mean: 253.4 days; range 22–473 days). Case 6 did not follow HIV treatment and was initially prescribed itraconazole for cutaneous sporotrichosis and died at another hospital without treatment with AMB. Ten (58.8%) patients received oral or parenteral steroids to modulate CNS inflammatory reaction. Fifteen (88.2%) patients had more than one admission for treatment of recurrence of neurological symptoms (1–8 admissions; mean 2.6), and the mean total length of hospital stay for treatment of meningitis was 150.6 days (0–577 days). Three (17.6%) patients underwent ventricular shunting to treat persistent hydrocephalus.

At the time of analysis, two patients were still on treatment with AMB. There was no significant difference in deaths among patients who had used steroids (72.2% × 33.3%; *p*: 0.162) or posaconazole (50% × 72.2%; *p*: 0.6) as well as those who had confirmation of CNS sporotrichosis by PCR or culture (63.3% × 36.3%; *p*: 0.17). Eleven (64.7%) patients died. Three (17.6%) patients are cured, and one (5.9%) patient is asymptomatic but with persistent CSF abnormalities after seven months of treatment.

## 4. Discussion

*Sporothrix brasiliensis* is hyperendemic in our region and highly virulent [13]. In a murine model, low fungal inoculum could be fatal and caused the extensive and early invasion of several organs including the CNS [3]. Central nervous disease is predominantly observed in patients with immunosuppression [8] and reports in immunocompetent individuals are rare [10,11,12,13,14]. Due to neurotropism, at INI-FIOCRUZ, all patients with the cutaneous disseminated form with immunosuppression and disseminated sporotrichosis undergo routine lumbar puncture even in the absence of neurological complaints. This approach allowed us to identify asymptomatic meningeal inflammation in more than 40% of the cases. In PLHIV, low-level meningeal inflammation is frequently observed during infection [15]. However, in this study, all patients with pleocytosis eventually developed clinical symptoms and signs consistent with chronic meningitis and had a partial or complete response to antifungal treatment. Since treatment may result in better clinical outcomes, we suggest that all patients with disseminated sporotrichosis due to *S. brasiliensis* undergo routine CSF evaluation.

Clinical symptoms and signs of CNS sporotrichosis are indistinguishable from other etiologies of chronic meningitis, but all patients herein described had overt disseminated cutaneous lesions at the time of diagnosis, which is a helpful clue. However, since immunosuppression is the leading risk factor for meningeal sporotrichosis, the careful assessment of more prevalent causes of chronic meningitis in this setting is mandatory. CSF analyses revealed mild to moderate pleocytosis associated with elevated protein and reduced glucose levels, which is frequently observed in patients with chronic meningitis irrespective of etiology [16]. Meningeal sporotrichosis is confirmed by a positive CSF culture, but *Sporothrix* spp. isolation in CSF is uncommon [6,8]. Presumptive diagnosis relies on symptoms and signs of chronic meningitis and confirmation of sporotrichosis from other sites [17]. In our study, less than half of the patients had positive fungal CSF culture. The use of CSF PCR allowed confirmation in additional cases with the advantage of being faster than culture, but further studies are necessary to determine the sensitivity and specificity of this technique [12]. Although serological testing of the CSF for antibodies to *Sporothrix* sp. can aid in the diagnosis, it is limited to the research setting and not commercially available [6,18].

Imaging findings are nonspecific and may be absent in some patients. Contrast enhancement of cerebral sulci and the basal cistern is highly suggestive of leptomeningitis, but it is not present in all cases. Blocking of CSF circulation led to ventricular dilatation in half of the patients and may be life-threatening, requiring ventricular shunting. Vascular lesions have been described previously in patients with meningeal sporotrichosis [19,20]. Donabedian et al. observed thrombotic endarteritis of small- and medium-sized brain vessels leading to multiple infarcts suggesting an inflammatory mechanism [19].

IRIS is a phenomenon occurring after the initiation of ART in HIV-infected individuals. A CD4+ T cell response towards an occult or a previously known opportunistic infection is the most accepted theory [21], and it usually occurs in the setting of a rapid rise in the CD4+ T cell count after profound depletion [22]. The incidence of CNS-related IRIS is variable, depending on the opportunistic infections implicated. In a Spanish study with 110 patients, the incidence of IRIS was 16.4% [23], while Dai et al. observed an incidence of 11.8% in 620 Chinese patients [24]. Although the occurrence of CNS IRIS in patients with meningeal sporotrichosis has been previously reported [8], in the present study, almost half of PLHIV with meningeal sporotrichosis developed new neurological symptoms or paradoxical worsening soon after ART. The reason for this phenomenon remains unknown, but since CNS IRIS can be potentially lethal, even with no evidence-based information, clinicians should consider delaying initiation of ART by a few weeks, similar to that recommended for cryptococcal and tuberculosis meningitis [21].

AMB is the available drug for the treatment of CNS sporotrichosis. Lipid-based formulations are preferred due to fewer adverse effects. Guidelines suggest Amphotericin B, given as a lipid formulation at a dosage of 5 mg/kg daily for 4–6 weeks followed by itraconazole 200 mg twice daily for 12 months [25,26]. Amphotericin B deoxycholate, administered at a dosage of 0.7–1.0 mg/kg daily, could also be used but was not preferred by the panel of experts. Nevertheless, most of our patients underwent longer courses of antifungal treatment, and readmissions due to relapses of meningitis were common, showing the difficulty of achieving persistent sterilization of the meningeal compartment. Direct AMB administration through intrathecal route is a viable option for fungal infections [27,28]; however, except for one case report [29], it has not been used in meningeal sporotrichosis.

Posaconazole has good activity against *Sporothrix* sp. [30,31]. However, its penetration into the CNS is very low [32]. Some studies on other fungal infections of the CNS demonstrate a good clinical response to its use, even with low, erratic, or undetectable CSF levels [33,34,35]. One possible explanation is that meningitis increases the permeability of the blood-brain barrier, contributing to the drug’s major effect. In the present cohort, in one patient with persistent fungal CSF isolation despite AMB, posaconazole resulted in a better clinical response and clearing of the fungus in the CSF [36]. Since then, it has been used in nearly all severe sporotrichosis cases at INI-FIOCRUZ.

There is also evidence of a synergistic effect when used with AMB in a murine model [37], but a clear benefit of this combination was not observed in our patients. Only a small group received posaconazole since it is not readily available for use in Brazil, and there was a bias toward its use in more severe cases as rescue therapy.

The role of steroids in the treatment of meningeal sporotrichosis is controversial. Like other IRIS-related CNS infections, corticosteroids may be helpful to HIV-infected patients who present with new neurological symptoms or have paradoxical worsening of previous diagnosed meningeal sporotrichosis upon introduction of ART [21]. In the present study, albeit not significant, there was a trend toward increased mortality in the group who received steroids. Possible explanations are selection bias, as those with a worse prognosis due to more severe CNS disease receive steroidal treatment to reduce or prevent the complications associated with meningeal inflammation. Alternatively, empirical use of steroids and antituberculosis drugs, commonly attempted in patients with chronic meningitis whose initial investigations are nonrevealing, may delay the correct diagnosis due to the transitory relief of neurological symptoms. In our experience, steroids should be used only in IRIS-associated meningeal sporotrichosis with neurological worsening despite antifungal treatment.

Since immunosuppression is the leading risk factor for meningeal sporotrichosis, restoring the host immune response is a determinant for survival in these patients [16]. Despite advances in diagnosis and neurocritical-care supportive measures, meningeal sporotrichosis is still associated with high lethality. Galhardo et al. reviewed 21 previously reported cases in 2010 and estimated the lethality to be 57% [8], similar to that observed in the present study. Indeed, patients who developed meningitis had a significantly higher chance of death than patients with disseminated sporotrichosis and no neurological complications, emphasizing the need for prompt diagnosis and better treatment regimens.

## 5. Conclusions

*S. brasiliensis* can be neurotropic, and CNS disease occurs predominantly in patients with disseminated sporotrichosis and immunosuppression. Since meningeal involvement is associated with a higher risk of death and can be initially asymptomatic, these patients should undergo lumbar puncture despite the absence of neurological symptoms. Better treatment strategies are needed since CSF sterilization is difficult to achieve with intravenous AMB alone, and relapses are common. The effect of association with posaconazole remains to be elucidated, but it may be an option for refractory cases. CNS sporotrichosis can be a manifestation of IRIS in PLHIV, and ART should be delayed for some weeks until improvement is achieved with antifungal treatment. Regular use of steroids is discouraged and should be reserved for patients with IRIS-associated disease and neurological deterioration.

## Figures and Tables

**Figure 1 jof-09-00017-f001:**
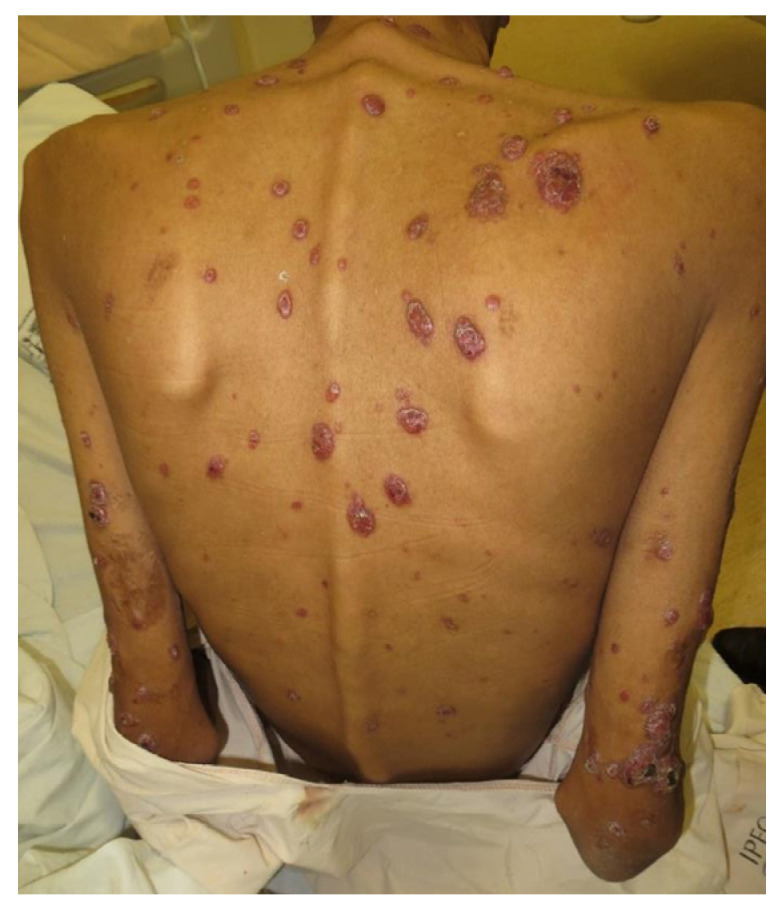
Disseminated cutaneous sporotrichosis: nodular ulcerative lesions on the trunk and arms.

**Figure 2 jof-09-00017-f002:**
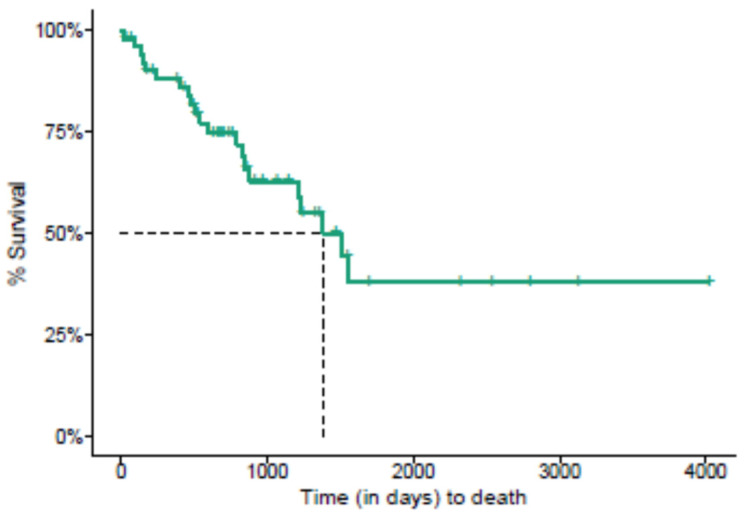
Kaplan–Meier survival analysis of the time to death in 53 patients with disseminated sporotrichosis at INI/FIOCRUZ.

**Figure 3 jof-09-00017-f003:**
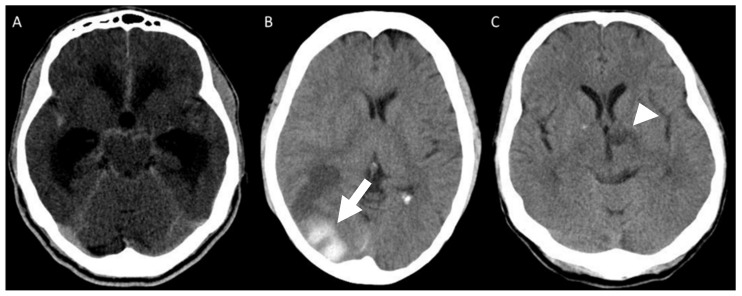
Brain CT scan showing (**A**) ventricular dilatation and basal cisterns contrast enhancement; (**B**) a parieto-occipital hemorrhagic lesion (white arrow) and (**C**) left thalamic ischemic lesion (white arrowhead).

**Table 1 jof-09-00017-t001:** Epidemiological, clinical, and laboratory findings of 53 patients with disseminated sporotrichosis at INI/FIOCRUZ.

	Group 1 (*n* = 17)	Group 2 (*n* = 36)
Mean age in years (range)	35.3 (20–57)	43 (16–72)
Male—*n* (%)	15 (88.2%)	29 (80.6%)
Non-white skin color—*n* (%)	16 (94.1%)	24 (66.7%)
Close contact with cats—*n* (%)	12 (70.6 %)	28 (77.8%)
Immunosuppression—*n* (%)		
HIV infection	15 (88.2%)	27 (75%)
Alcoholism	1 (5.9%)	2 (5.6%)
Chronic steroid use	1 (5.9%)	4 (11.1%)
None of the above	-	3 (8.3%)
Mean duration of signs/symptoms until lumbar puncture in months (range)	6.9 (1–20)	7.4 (1–67)
Mean CSF cells/mm^3^ (range)	71.5 (3–363)	2 (0–5)
Mean CSF glucose mg/dL (range)	39.6 (14–65)	59.1 (27–148)
Mean CSF protein mg/dL (range)	117 (32.3–307)	50.1 (13–181)
Mean duration of first hospital admission in days (range)	45.1 (10–120)	28.2 (1–86)
ICU admission—*n* (%)	6 (35.3%)	0
Death—*n* (%)	11 (64.7%)	10 (27.8%)

Group 1: patients with symptoms and signs of meningitis; group 2: patients with no neurological symptoms and CSF cell count ≤ 5 cells/mm³. HIV: human immunodeficiency virus. CSF: cerebrospinal fluid. ICU: intensive care unit.

**Table 2 jof-09-00017-t002:** Crude and Adjusted Hazard Ratios (HR) by Cox regression models for overall survival in 53 patients with disseminated sporotrichosis at INI/FIOCRUZ.

	Levels	*n* (%)	HR (Crude)	HR (Adjusted)
Sex	Female Male	9 (17) 44 (83)	0.72 (0.2–2.58, *p* = 0.609)	0.79 (0.21–3, *p* = 0.733)
Skin color	White Non-white	13 (24.5) 40 (75.5)	1.7 (0.49–5.98, *p* = 0.406)	0.95 (0.23–3.87, *p* = 0.945)
Age	Mean (SD)	40.5 (12.5)	1 (0.95–1.04, *p* = 0.92)	1.02 (0.96–1.07, *p* = 0.548)
Time of onset	≤3 months >3 months	22 (41.5) 31 (58.5)	1.27 (0.5–3.22, *p* = 0.608)	0.72 (0.25–2.11, *p* = 0.554)
Meningitis	No Yes	36 (67.9) 17 (32.1)	3 (1.24–7.26, *p* = 0.015)	3.87 (1.23–12.17, *p* = 0.021)
HIV infection	No Yes	11 (20.8) 42 (79.2)	0.55 (0.17–1.79, *p* = 0.323)	0.67 (0.2–2.29, *p* = 0.521)

HIV: human immunodeficiency virus; SD: standard deviation; HR: hazard ratio.

**Table 3 jof-09-00017-t003:** Characteristics of patients with central nervous system sporotrichosis (Group 1).

Patient	Sex/Age ^a^ (yo)	Predisposing Condition	Duration of Neurological Symptoms	Clinical Manifestations	Initial CSF Findings	CSF *Sporothrix* spp. Detection ^b^	Radiological Features	Treatment	Outcome
1	M/44	HIV infection	15 days	Headache, right III nerve palsy, cognitive complaints	363 cells/mm^3^ (91%MN), P: 410 mg/dL; G: 31 mg/dL	Positive *Sporothrix* spp. PCR	Left thalamus ischemic stroke	AmB + posaconazole	Deceased
2	M/57	HIV infection	1 month	Headache	90 cells/mm^3^ (100%MN), P: 63.9 mg/dL; G:48 mg/dL	No	None	AmB + posaconazole	Ongoing treatment (AmB)
3	M/28	HIV infection	2 months	Headache, vomiting, nuchal rigidity	206 cells/mm^3^ (90%MN), P: 159 mg/dL; G: 24 mg/dL	No	Meningeal contrast enhancement	AmB	Cured
4	M/46	HIV infection	4 months	Headache, vomiting, nuchal rigidity, cognitive complaints	12 cells/mm^3^ (86%MN), P: 293 mg/dL; G: 22 mg/dL	No	Hydrocephalus, right internal capsule ischemic stroke	AmB	Deceased
5	M/26	HIV infection	15 days	Headache, nuchal rigidity, lethargy, seizures	6 cells/mm^3^ (100%MN), P: 307 mg/dL; G: 27 mg/dL	Positive culture	Hydrocephalus, meningeal contrast enhancement, and nodular lesion in the right cerebellar hemisphere	AmB	Deceased
6	M/37	HIV infection	1 month	Headache, seizures	10 cells/mm^3^ (100%MN), P: 97.5 mg/dL; G: 52 mg/dL	No	None	Itraconazole	Deceased
7	M/41	Alcoholism	1 month	Drowsiness, nuchal rigidity	79 cells/mm^3^ (78%MN), P: 162 mg/dL; G: 32 mg/dL	No	None	AmB	Deceased
8	M/48	HIV infection	15 days	Headache, vomiting, seizures	4 cells/mm^3^ (98%MN), P: 61.6 mg/dL; G: 35 mg/dL	Positive culture	Hydrocephalus	AmB	Deceased
9	F/21	HIV infection	1 month	Headache, right VI nerve palsy, drowsiness, nuchal rigidity	3 cells/mm^3^ (100%MN), P: 33 mg/dL; G: 59 mg/dL	Positive culture	Hydrocephalus, meningeal contrast enhancement	AmB + posaconazole	Deceased
10	M/40	HIV infection	2 months	Headache, vomiting, drowsiness, cognitive complaints	12 cells/mm^3^ (97%MN), P: 32 mg/dL; G: 65 mg/dL	No	Hydrocephalus, meningeal contrast enhancement	AmB	Deceased
11	M/28	Leprosy and chronic steroid use	2 months	Headache	133 cells/mm^3^ (37%MN), P: 278 mg/dL; G: 58 mg/dL	Positive culture	None	AmB	Asymptomatic but with persistent CSF abnormalities
12	F/31	HIV infection	1 month	Headache, vomiting, motor deficit, seizures, drowsiness	25 cells/mm^3^ (90%MN), P: 61 mg/dL; G: 10 mg/dL	Positive *Sporothrix* spp. PCR	Hydrocephalus, meningeal, contrast enhancement, right parietal hemorrhagic stroke	AmB + posaconazole	Deceased
13	M/38	HIV infection	1 month	Vomiting, cognitive complaints	86 cells/mm^3^ (64%MN), P: 74.2 mg/dL; G: 35 mg/dL	Positive culture	Hydrocephalus	AmB	Deceased
14	M/43	HIV infection	8 months	Headache, vomiting, VI nerve palsy, drowsiness	15 cells/mm^3^ (100%MN), P: 47.5 mg/dL; G: 53 mg/dL	Positive *Sporothrix* spp. PCR	Hydrocephalus, meningeal contrast enhancement	AmB	Cured
15	M/23	HIV infection	15 months	Headache, vomiting, nuchal rigidity, drowsiness	72 cells/mm^3^ (100%MN), P: 100 mg/dL; G: 34 mg/dL	Positive *Sporothrix* spp. PCR	Hydrocephalus, meningeal contrast enhancement	AmB + posaconazole	Cured
16	M/35	HIV infection	2 months	Headache, vomiting, drowsiness	5 cells/mm^3^ (100%MN), P: 42.5 mg/dL; G: 57 mg/dL	Positive culture	None	AmB	Deceased
17	M/20	HIV infection	3 months	Headache	86 cells/mm^3^ (75%MN), P: 68.6 mg/dL; G: 35 mg/dL	Positive culture	Meningeal contrast enhancement	AmB + posaconazole	Ongoing treatment (AmB)

^a^ Age at onset; ^b^
*Sporothrix* spp. PCR reported by Oliveira et al., 2020 [12]. Yo: years-old; M: male; F: female; CSF: cerebrospinal fluid; P: proteins; G: glucose; PCR: polymerase chain reaction; MN: mononuclear cells; AmB: amphotericin B.

## Data Availability

All relevant data are presented in the article.

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
