# Peer review of "Meningeal Sporotrichosis Due to Sporothrix brasiliensis: A 21-Year Cohort Study from a Brazilian Reference Center"

_jof, 2022, doi:10.3390/jof9010017_

Round 1

Reviewer 1 Report

This is a useful paper reporting on sporotrichosis, an endemic and serious disease in Brazil and other countries. The authors investigate parameters associated with meningeal sporotrichosis compared to non-neurological disease, and present a number of interesting findings. The paper is clear and well written, with just a few errors of English that I have noted in the marked up copy attached. It will be of interest to researchers and clinicians that deal with invasive mycoses. 

Author Response

Dear reviewer #1

Thanks for the comments. All the queries and errors were answered and corrected throughout the text. 

Reviewer 2 Report

Comments to authors:

The manuscript is clear, concise and provides information about diagnosis and treatment of meningeal sporotrichosis.

In addition, authors highlighted the importance of screening LP in patients with disseminated disease despite the absence of neurological complaints.

This article should be useful not only for clinical laboratory practitioners, but also for clinical physicians.

 Only a few corrections, please delete the period at the end of the title.

I have a question regarding the use of itraconazole as treatment (patient 6). Taking into account the PK/PD of itraconazole, could you explain or justify the election of itraconazole for treatment?

Author Response

Dear reviewer #2

Thanks for the careful review.

The period at the end of the title was removed as requested.

The patient #6 received itraconazole for the treatment of cutaneous sporotrichosis. He was admitted at another hospital for treatment of meningitis and died before starting AMB. This information was added in the RESULTS section.